# Clinical and Metabolic Characteristics of Hyperuricemia with Risk of Liver Fibrosis: A Cross-Sectional Study

**DOI:** 10.3390/metabo12100893

**Published:** 2022-09-23

**Authors:** Chun-Yi Wang, Hsiang-Han Kao, Kuan-Yu Lai, Ching-Chun Lin, Wen-Yuan Lin, Chiu-Shong Liu, Tsung-Po Chen

**Affiliations:** 1Department of Community and Family Medicine, China Medical University Hospital, Taichung 404, Taiwan; 2Department of Social Medicine and Family Medicine, College of Medicine, China Medical University, Taichung 404, Taiwan

**Keywords:** non-alcoholic fatty liver disease, advanced fibrosis, hyperuricemia, type 2 diabetes

## Abstract

The role of serum uric acid (SUA) in the role of advanced fibrosis is not fully explored. The study assesses the risk of advanced fibrosis according to SUA in an Asian population with a total of 3612 subjects enrolled in one health management center between 2006 and 2008. The fibrosis-4 score was used for the prediction of the high risk of advanced fibrosis. SUA scores higher than 7.6 mg/dL in men and 6.6 mg/dL in women were defined as hyperuricemia. A proportional odds model was used to assess cumulative risks of advanced fibrosis. The prevalence of high risk of advanced fibrosis was 2.5% in the hyperuricemia group and 0.6% in the normal SUA group (*p* < 0.001). After adjustment for confounding factors, the odds ratios (OR) for more severe advanced fibrosis were 1.37 (95% confidence interval [CI]: 1.07–1.78) in the hyperuricemia group. Hyperuricemia only increased the risk of advanced fibrosis in the non-T2DM group (OR, 1.29; 95% CI, 1.04 to 1.74) instead of T2DM group (OR, 1.85; 95% CI, 0.97 to 3.53). SUA is a risk factor for a higher risk of advanced fibrosis, with the disease likely progressing from a steatotic to a fibrotic picture. The focus should be more emphasized in non-T2DM groups.

## 1. Introduction

Non-alcoholic fatty liver disease (NAFLD) is one of the most common liver diseases and the prevalence has been rising in recent decades worldwide [1,2]. It is estimated that NAFLD has a global prevalence of 25%. The progression of NAFLD ranges from simple steatosis, to non-alcoholic steatohepatitis (NASH), to hepatic fibrosis, to hepatocellular carcinoma (HCC) [2,3]. NAFLD will become the main cause of HCC after the global campaign to eliminate the hepatitis B and hepatitis C virus infections. The efforts to decrease NAFLD progression to NASH and advanced fibrosis is an important issue for preventing future HCC.

The dynamic mechanism of liver fibrinogenesis proceeds by steps such as the proinflammatory stage, perpetuation, and followed by resolution phase [4,5,6]. The role of uric acid in serum as an oxidation product of purine metabolism with inflammatory cause and association of cardiovascular disease, insulin resistance, renal disease, and metabolic syndrome was considered to be connected to liver fibrosis [7]. In the past, the confirmed diagnosis of liver fibrosis was through biopsy, which is often underdiagnosed in routine practice [8]. Additionally, the invasive nature of liver biopsy is seldom used in the epidemiologic survey but in the definite diagnosis. Therefore, the development of non-invasive tools with good diagnostic accuracy is important for the early screening and prediction of advanced fibrosis in the NAFLD group.

The technique of the liver fibrosis approach in a conservative way includes serum markers such as serum albumin, fibrotest, aspartate aminotransferase (AST) to platelet ratio index (APRI), FIB-4 index, AST to alanine aminotransferase (ALT) ratio, fibrosis-4 index, BARD score, NAFLD fibrosis score (NFS), or modalities such as acoustic radiation force impulse (ARFI), transient elastography (TE), magnetic resonance elastography (MRE), shear wave elastography (SWE), and fibroscan [9,10,11,12]. Xiao et al. collected 64 articles with 13,046 NAFLD subjects involved to compare the diagnostic accuracy of non-invasive tests for staging fibrosis in NAFLD. The BARD score, NFS, and fibroscan were found sensitive for the evaluation of the general population [13]. Additionally, there is increasing ratio of NAFLD patients with connection of type 2 diabetes mellitus (T2DM) and further, T2DM patients who later develop into NAFLD [14,15]. Xu et al. found patients with T2DM with elevation of serum uric acid level is an independent risk factor for the prevalence of NAFLD [16,17].

Due to the above, we approach the association between uric acid level with advanced fibrosis with the effect of T2DM and sex comparison in Taiwan.

## 2. Materials and Methods

### 2.1. Subjects

Participants were recruited from China Medical University Hospital health management center for their health survey during the period 2006 to 2008. The details of study protocols were described in our previous study [18]. The exclusion criteria were previous history of chronic liver diseases (including chronic hepatitis, autoimmune, drug-induced, vascular and inherited hemochromatosis, and Wilson disease), or excessive alcohol use (>10 g of alcohol daily for women and >20 g for men). Subjects using anti-hyperuricemic agents were excluded from our study. Furthermore, we excluded those who were aged equal to or older than 65 years.

### 2.2. Data Collection

Information about general characteristics and lifestyles was collected by experienced investigators. All participants completed detailed surveys and basic demographic data, medical history, and lifestyle characteristics, including age, sex, past medical history, cigarette smoking, alcohol drinking, and exercise habits. The T2DM, hypertension, and dyslipidemia statuses were recorded based on the questionnaire. Anthropometric measurements were recorded. Body mass index (BMI) was calculated with the standard formula: weight (kg)/(height [m])^2^. Blood pressure was measured twice with an automated sphygmomanometer. Pasting plasma glucose (FPG), liver function test including aspartate aminotransferase (AST) and alanine aminotransferase (ALT), gamma-glutamyl transpeptidase (r-GT), and platelet count, total cholesterol (Total-C), triglyceride, high-density lipoprotein cholesterol (HDL-C), low-density lipoprotein cholesterol (LDL-C), and uric acid were measured after a 12 h overnight fast.

### 2.3. Definition of Hyperuricemia

Hyperuricemia was defined as serum uric acid >7.6 mg/dL in men and >6.6 mg/dL in women. Furthermore, subjects with past history of hyperuricemia and taking anti-hyperuricemic medication were classified as hyperuricemia group.

### 2.4. Diagnosis of Type 2 Diabetes

The diagnostic criteria for T2DM were based on the American Diabetes Association guidelines [19] as one of the following factors: (1) history of T2DM and use of anti-diabetic agent currently, (2) fasting plasma glucose ≥126 mg/dL, (3) hemoglobin A1c ≥6.5%.

### 2.5. Diagnosis of Chronic Kideny Disease

The diagnostic criteria for chronic kidney disease (CKD) were based on (1) history of CKD in the questionnaire, (2) estimated glomerular filtration rate (GFR) less than 60 mL/min/1.73 m^2^.

### 2.6. Prediction of High Risk of Advanced Fibrosis

The FIB-4 index was calculated as [age (years) × AST (U/L)]/[platelets (10^9^/L) × ALT (U/L)^1/2^] [20,21]. The subjects were classified into three groups: high risk FIB-4, indeterminate risk FIB-4, and low risk FIB-4. The cutoff value was based on the following values: FIB-4 index ≥ 2.67, 1.45 ≤ FIB-4 < 2.67, and <1.45.

### 2.7. Statistical Analysis

Data are summarized as the mean ± standard error for continuous variables and as the percentage for categorical variables. Statistical analysis was performed using Student’s *t*-test for continuous variables and the chi-squared test for categorical variables. The cumulative risk of more severe advanced fibrosis (the probability of being indeterminate-to-high versus low and the probability of being high versus being low-to-indeterminate) was estimated by odds ratios (ORs) and 95% confidence intervals (CIs). We checked risk factors for developing advanced fibrosis with sex, BMI, T2DM, CKD, exercise, smoking and drinking habits. Model 1 showed the crude odds ratio of advanced fibrosis between two groups. Model 2 added the age, sex, BMI, T2DM and CKD of the participants as confounding factors, while model 3 furtherly included exercise habit and social factors, including alcohol intake and smoking. A two-tailed *p*-value < 0.05 was considered significant. All analyses were conducted in SAS version 9.4 (SAS Inc., Cary, NC, USA).

## 3. Results

A total of 3612 participants were involved in the study from 2006 to 2008. The anthropometric and laboratory markers between hyperuricemia and normal groups are shown in Table 1. The mean level of SUA were 8.3 ± 1.0 mg/dL in the hyperuricemia group and 5.5 ± 1.1 mg/dL in the normal group. The prevalence of hyperuricemia was 14.8%. The proportions of subjects diagnosed with advanced fibrosis from low, indeterminate, and high were 77.4%, 20.1%, and 2.5% in hyperuricemia group, respectively, and 83.2%, 16.2%, and 0.6% in normal group, respectively (*p* < 0.05). The hyperuricemia group had older age and higher percentage of men, and a higher percentage of CKD than the normal group. Compared with the normal group, the hyperuricemia group had higher weight, waist circumference, BMI, SBP, DBP, FPG, AST, ALT, r-GT, triglyceride, and LDL-C levels but lower HDL-C and estimated GFR levels (*p* < 0.05).

Table 2 shows the predictors of high risk of advanced fibrosis defined by FIB-4 analyzed with cumulative logistic regression model. Comparing with normal SUA level group, the increased risk of advanced fibrosis in hyperuricemia group was significant after adjusting for confounding factors (OR, 1.37; 95% CI, 1.07 to 1.78). Furthermore, those with diagnosis of T2DM (OR, 2.20; 95% CI, 1.67 to 2.91) and CKD (OR, 2.43; 95% CI, 1.53 to 3.86) had higher risk for advanced fibrosis. In addition, the risk of advanced fibrosis was significantly lower in women than in men (OR, 0.75; 95% CI, 0.61 to 0.92).

In the analysis stratified by men and women in Table 3, hyperuricemia increases risk of advanced fibrosis in women (OR, 1.71; 95% CI, 1.08 to 2.70) but not in men (OR, 1.26; 95% CI, 0.94 to 1.68). We furtherly stratified with T2DM and non-T2DM groups in Table 4. Hyperuricemia only increased risk in the non-T2DM group (OR, 1.29; 95% CI, 1.04 to 1.74) but not in the T2DM group (OR, 1.85; 95% CI, 0.97 to 3.53).

## 4. Discussion

Our study has the following novel findings: hyperuricemia has an increased risk of advanced fibrosis. Women have a lower risk of advanced fibrosis compared to men; however, women with hyperuricemia are at a higher risk for advanced fibrosis than men. Hyperuricemia is a risk factor for advanced fibrosis in those without diabetes compared with those with T2DM.

NAFLD is the most prevalent chronic liver disease worldwide and the main cause of hepatic cancer after the eradication of hepatitis virus [22,23]. NAFLD progress to NASH with feature of hepatocellular ballooning and lobular inflammation [24]. NASH has higher risk for advanced fibrosis and HCC compared with simple steatosis alone [25]. Effective screening in the selected cohorts and community is mandatory to define treatment strategies and early identification of risk factors. Uric acid is one of the metabolic abnormalities. In 2002, Lonardo et al. reported that the elevated SUA level was positively associated with NAFLD in an Italian population [26]. SUA as a risk factors for NAFLD was established in the following studies. More recently, a meta-analysis with 55,573 subjects indicated that the level of SUA was still associated to NAFLD after adjusting sex, age, and MetS [27]. Early identification of those at risk is needed in timely discussions about the disease, screening, and prevention, especially through lifestyle modifications. However, the link between hyperuricemia and advanced fibrosis is still under debate.

Afzali et al. demonstrates that the serum SUA level is associated with the cirrhosis and the elevated serum liver enzymes after adjustments for important causes and risk factors of chronic liver disease [28]. In contrast, a previous cross-sectional study from Taiwan that included 130 patients has suggested a significant inverse relationship between SUA and fibrosis stages with biopsy-proven NAFLD [29]. With a conflicting result, Jaruvongvanich et al. conducted a meta-analysis study in the NAFLD group. Patients with hyperuricemia were not significantly associated with advanced fibrosis in five observational studies including total of 749 participants [30]. The results might not be robust due to few participants included even in meta-analysis study. Furthermore, most studies included in the meta-analysis are cross-sectional studies. The liver biopsy is a gold standard for the diagnosis of NAFLD; however, due to invasive procedures and possible side effects, it is seldom used in the epidemiologic study. Instead, many non-invasive tests (NITs) are used for survey. Recent published study by using APRI shows the positive association between hyperuricemia and significant fibrosis in NAFLD but not in the non-NAFLD group [8]. Compared with the study, we use FIB-4, which is the most frequently used method for assessing hepatic fibrosis, instead of APRI. NIT has good diagnostic power in previous studies. Preliminary study has demonstrated that the FIB-4 score was a surrogate marker of hepatic fibrosis, providing high diagnostic accuracy for advanced fibrosis with an area under an ROC curve (AUROC) of 0.86 [31]. The strength of our study is the first study to examine the association between SUA and advanced fibrosis by using FIB-4 and finding a positive association. Compared with previous studies, more participants were involved in our study. Furthermore, NIT is suitable for epidemiologic study and early findings for risk groups for referral.

T2DM is a risk factor for NAFLD and advanced fibrosis [14]. Approximate one out of six T2DM patients has at least moderate-to-advanced fibrosis. A group of experts has called for action for systematic screening for NASH [32] and liver fibrosis [14] in the outpatient setting. Our study also demonstrates that T2DM is a risk for advanced fibrosis. However, the role of SUA is not consistent in those with T2DM and without T2DM. This inconsistency might derive from the role of interaction between SUA and T2DM through systemic inflammation and insulin resistance [33]. In those without T2DM, hyperuricemia independently affects the NAFLD and advanced fibrosis, and the later result is shown in our study.

Sex differences exist in the prevalence, risk factors, fibrosis, and clinical outcomes of NAFLD [34]. Our results show that women pose a lower risk for advanced fibrosis than men. This positive association is also demonstrated in previous studies [35,36,37]. The role of SUA is different in men and women groups. Hyperuricemia increase risk of advanced fibrosis only in women group but not in the men’s group. This result might come from the potential sex-specific effects or hormonal effects. Further studies are needed for answering the question.

Exercise improved hepatic steatosis underlying metabolic abnormalities and liver fibrosis [38,39,40]. Exercise alone improved intrahepatic triglyceride levels [41]. Furthermore, exercise combined with dietary intervention improves serum levels of liver enzymes and liver fat or histology [39]. However, Table 4 demonstrates exercise was a significant risk factor for advanced liver fibrosis in non-diabetic subjects (OR, 1.92; 95% OR, 1.55–2.37). Whether this result is really clinically significant or just statistically significant is not explored. This result might result from possible interaction of insulin resistance and hepatic steatosis with exercise. Wang et al. demonstrated that physical activity alone can only slightly improve hepatic enzyme levels and intra-hepatic lipid content in non-diabetic patients with NAFLD [42]. Additionally, exercise type and intensity might also influence the study result and further studies in specific groups are needed.

Our study highlights the need for NAFLD and advanced fibrosis surveys in the hyperuricemia population. The presence of fibrosis, in particularly advanced fibrosis (stage 3 and 4), is an important prognostic marker for liver-related outcomes and overall mortality [43]. The cost-effectiveness of NAFLD screening and who to screen is still under debate [44]. Thus, screening in the group of patients with these risk factors for complications is particularly important. The European Association for the Study of the Liver (EASL) proposed two-stage screening combing primary care/diabetology clinic and liver clinic for risk identification [45]. The American Gastroenterology Association (AGA) also proposed NIT and VCTE for identifying advanced fibrosis risk group [46]. Thus, our study might contribute to risk identification of advanced fibrosis by using NIT for possible early referral.

Although the underlying biological mechanism linking SUA to advanced fibrosis is not entirely understood, several plausible mechanisms have been proposed. Oxidative stress and lipid peroxidation were the main causes of fatty liver. SUA was significantly associated with the degree of steatosis and the greater odds of advanced lobular inflammation of NAFLD. Additionally, hyperuricemia is a component of MetS; the increase in SUA levels could promote oxidative stress and reactive oxygen species levels, eventually increasing the risk for advanced fibrosis.

This study has several limitations to be considered in the interpretation of our findings. First, we cannot demonstrate a causal relationship between SUA and advanced because of the cross-sectional study design. Second, we used FIB-4 instead of a liver biopsy to assess advanced fibrosis. Liver biopsy is considered the gold standard for diagnosis. However, FIB-4 is widely used as the first step for identifying high risk group of advanced fibrosis. The efficacy for the diagnosis of advanced fibrosis is well established and recommended for the epidemiological study. Third, we did not adjust age as a confounding factor in our regression model. Due to age being a component of FIB-4, adjusting age would be a great bias. However, we limited the effect of age by focusing our group on ages under 65 years old, which makes our participants more homogeneous. Fourth, those who used fibrates for hypertriglycemia might improve advanced fibrosis status. However, we only recorded those who use anti-lipid drugs, which indicated statins in most cases. Thus, we cannot adjust the effect of fibrates in our study.

## 5. Conclusions

In conclusion, SUA carries an increased prevalence of high risk for advanced fibrosis. Early identification of risk groups among hyperuricemia populations should be highlighted.

## Figures and Tables

**Table 1 metabolites-12-00893-t001:** Baseline clinical and biochemical characteristics of the study subjects.

	Normal	Hyperuricemia	
	N = 3089	N = 523	
Men	1555 (50.3%)	379 (72.5%)	<0.0001
Age (years)	46.5 ± 9.8	48.7 ± 9.9	<0.0001
Smoking	21.7%	24.8%	0.12
Alcohol	27.6%	37.8%	<0.0001
Exercise	59.2%	60.9%	0.49
T2DM	8.4%	10.3%	0.15
CKD	1.8%	8.1%	<0.0001
Height	163.5 ± 8.0	165.7 ± 8.6	<0.0001
Weight (kg)	62.6 ± 11.7	71.0 ± 11.8	<0.0001
Waist (cm)	79.0 ± 9.9	87.2 ± 9.2	<0.0001
BMI (kg/cm^2^)	23.3 ± 3.5	25.8 ± 3.4	<0.0001
Systolic BP (mmHg)	116.6 ± 14.9	122.7 ± 14.8	<0.0001
Diastolic BP (mmHg)	74.8 ± 10.3	79.0 ± 10.0	<0.0001
FPG (mg/dL)	95.4 ± 25.9	98.1 ± 22.8	0.03
Total-C (mg/dL)	198.6 ± 38.0	211.9 ± 42.4	<0.0001
HDL-C (mg/dL)	45.3 ± 13.5	39.4 ± 11.0	<0.0001
LDL-C (mg/dL)	128.6 ± 35.3	140.5 ± 38.8	<0.0001
Triglyceride (mg/dL)	115.7 ± 85.0	168.2 ± 113.8	<0.0001
ALT (IU/L)	27.2 ± 18.1	50.0 ± 28.5	<0.0001
AST (IU/L)	24.7 ± 10.6	32.0 ± 20.7	<0.0001
r-GT (IU/L)	27.6 ± 41.0	50.7 ± 141	<0.0001
SUA (mg/dL)	5.5 ± 1.1	8.3 ± 1.0	<0.0001
Serum Cr (mg/dL)	0.86 ± 0.22	1.04 ± 0.61	<0.0001
eGFR	92.6 ± 18.7	80.6 ± 16.0	<0.0001
BUN (mg/dL)	11.5 ± 3.5	13.0 ± 8.1	<0.0001
Advanced fibrosis			<0.0001
Low	83.2%	77.4%	
Indeterminate	16.2%	20.1%	
High	0.6%	2.5%	

Abbreviation: T2DM, type 2 diabetes; CKD, chronic kidney disease; BMI, body mass index; BP, blood pressure; FPG, fasting plasma glucose; Total-C, total cholesterol; HDL-C, high-density lipoprotein cholesterol; LDL-C, low-density lipoprotein cholesterol; ALT, alanine transaminase; AST, aspartate transaminase; r-GT, gamma glutamyl transpeptidase; SUA, serum uric acid; Serum Cr, serum creatinine; eGFR, estimated glomerular filtration rate; BUN, blood urea nitrogen. Advanced fibrosis category: high risk, FIB-4 index ≥ 2.67; indeterminate, 1.45 ≤ FIB-4 < 2.67; low risk, FIB-4 index < 1.45.

**Table 2 metabolites-12-00893-t002:** Cumulative logistic regression model to identify the risk [odds ratio, OR (95% CI)] of factors for advanced fibrosis.

	Model 1	Model 2	Model 3
Uric acid (ref = normal)	1.66 (1.33–2.06)	1.38 (1.08–1.77)	1.37 (1.07–1.78)
Sex (ref = men)		0.79 (0.65–0.95)	0.75 (0.61–0.92)
BMI		1.00 (0.98–1.03)	1.00 (0.98–1.03)
T2DM		2.28 (1.72–3.00)	2.20 (1.67–2.91)
CKD		2.42 (1.53–3.84)	2.43 (1.53–3.86)
Smoking (ref = no)			0.75 (0.58–0.96)
Drinking (ref = no)			1.02 (0.82–1.26)
Exercise (ref = no)			1.73 (1.41–2.11)

Abbreviation: BMI, body mass index; T2DM, type 2 diabetes; CKD, chronic kidney disease. Model 1: crude; Model 2: adjust for sex, BMI, T2DM, CKD; Model 3: adjust for sex, BMI, T2DM, CKD, smoking, drinking, exercise.

**Table 3 metabolites-12-00893-t003:** Risk for advanced fibrosis according to sex group.

	Odds Ratio
Men	
Uric acid (ref = normal)	1.26 (0.94–1.68)
BMI	1.00 (0.96–1.04)
T2DM	1.94 (1.36–2.79)
CKD	2.78 (1.64–4.71)
Smoking (ref = no)	0.73 (0.56–0.95)
Drinking (ref = no)	1.05 (0.82–1.35)
Exercise (ref = no)	1.52 (1.17–1.96)
Women	
Uric acid (ref = normal)	1.71 (1.08–2.70)
BMI	1.00 (0.96–1.04)
T2DM	2.66 (1.70–4.15)
CKD	1.54 (0.57–4.21)
Smoking (ref = no)	0.87 (0.39–1.90)
Drinking (ref = no)	0.94 (0.57–1.54)
Exercise (ref = no)	2.06 (1.52–2.80)

Abbreviation: BMI, body mass index; T2DM, type 2 diabetes; CKD, chronic kidney disease.

**Table 4 metabolites-12-00893-t004:** Risk for advanced fibrosis according to type 2 diabetes.

	Odds Ratio
T2DM	
Uric acid (ref = normal)	1.85 (0.97–3.53)
Sex (ref = men)	0.79 (0.44–1.43)
BMI	1.01 (0.95–1.08)
CKD	0.87 (0.25–3.03)
Smoking (ref = no)	0.50 (0.24–1.05)
Drinking (ref = no)	0.82 (0.43–1.58)
Exercise (ref = no)	0.78 (0.45–1.35)
Non-T2DM	
Uric acid (ref = normal)	1.29 (1.04–1.74)
Sex (ref = men)	0.74 (0.59–0.93)
BMI	1.00 (0.97–1.03)
CKD	2.99 (1.81–4.93)
Smoking (ref = no)	0.78 (0.60–1.02)
Drinking (ref = no)	1.05 (0.83–1.33)
Exercise (ref = no)	1.92 (1.55–2.37)

Abbreviation: BMI, body mass index, T2DM, type 2 diabetes; CKD, chronic kidney disease.

## Data Availability

The data presented in this study are available on request from the corresponding author. The data are not publicly available due to possibility of participants identification.

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
