# Peer review of "Clinical and Metabolic Characteristics of Hyperuricemia with Risk of Liver Fibrosis: A Cross-Sectional Study"

_metabolites, 2022, doi:10.3390/metabo12100893_

Round 1

Reviewer 1 Report

Wang and collaborators present a well-structured investigation demonstrating in a fairly sized cohort that serum uric acid (SUA) can be considered as a risk factor for liver fibrosis, especially in non-type 2 diabetic individuals.

The submitted manuscript is in line with the scopes of the journal.

While the study is convincing and well performed, the data presented in table 1 show that quite a large number of variables are highly different between the “normal” and “hyperuricemia” groups. Indeed,

- sex composition

-age

- Smoking status

-alcohol use (within the eligibility limits)

- body parameters (height, weight,waist circumference and BMI)

- cardiac and analytical parameters (blood pressure, cholesterol, transaminases).

Are all statistically different between the two groups.

Although table 2 presents logistic regressions to adjust for these factors, the reader is left with a doubt that any of these factors (especially alcohol use, or even the difference in age), may likewise statistically contribute to the instalment of liver fibrosis as SUA may do. I believe that this point should be clearly considered in the discussion section. Especially, (as stated in line 230 as a limitation of this study), adjusting for age would contribute to clarify the overall findings of the manuscript. 

Minor points:

Line 9: instead of “inconsistent”, I would rather use the concept of “not fully explored”

Paragraph 2.5: please amend for line spacing. The same applies to Paragraph “5. Conclusion”

Line 255 and 227: authors probably mean “advanced fibrosis”.

Line 249: The data availability statement could be limited to the second period. (i.e.: “Data available on request due to restrictions eg privacy or ethical.”  can be removed).

Author Response

Question 1: While the study is convincing and well performed, the data presented in table 1 show that quite a large number of variables are highly different between the “normal” and “hyperuricemia” groups. Indeed, sex composition, age, Smoking status, alcohol use (within the eligibility limits), body parameters (height, weight,waist circumference and BMI), cardiac and analytical parameters (blood pressure, cholesterol, transaminases). Are all statistically different between the two groups.

Answer 1: We appreciate your comments. In table 1, we use Student’s t-test for continuous variables and the chi-squared test for categorical variables. The results show statistically significant between normal and hyperuricemia groups. Highly difference between two groups might result from large number of participants in our study, which might make p-value more significant. However, it also means these variables are really different between two groups after statistical test. Whether these statistically significant differences really important to clinical practice might base on health practitioners’ daily practice.

Question 2: Although table 2 presents logistic regressions to adjust for these factors, the reader is left with a doubt that any of these factors (especially alcohol use, or even the difference in age), may likewise statistically contribute to the instalment of liver fibrosis as SUA may do. I believe that this point should be clearly considered in the discussion section. Especially, (as stated in line 230 as a limitation of this study), adjusting for age would contribute to clarify the overall findings of the manuscript.

Answer 2: We appreciate to your comment. Advanced fibrosis is a complex mechanism, resulting from simple hepatic steatosis, steatohepatitis. We select these factors were based on previous studies, or those would contribute to NAFLD or NASH. As mention to alcohol use, Blomdahl et al. demonstrated that moderate alcohol consumption was associated with advanced fibrosis in NAFLD population.[1] Age might be one of confounding factor of advanced fibrosis; however, age is one of variables in fibrosis-4, including age, AST, ALT and platelet. If we adjust age, there would be no effect obviously. McPherson et al. also suggested that adjusting cutoff value of FIB-4 was needed for those age older than 65 years.[2] For minimizing possible effect of age, we limit our participants less than 65 years old for more homogenous distribution between two groups. We discuss the limitation of age effect in our limitation part.

Question 3: Line 9: instead of “inconsistent”, I would rather use the concept of “not fully explored”

Answer 3: We amend the description as follow:

Line 9: “The role of serum uric acid (SUA) in the role of advanced fibrosis is not fully explored”

Question 4: Paragraph 2.5: please amend for line spacing. The same applies to Paragraph “5. Conclusion”

Answer 4: We amend for line spacing in the paragraph 2.5 and 5.

Question 5: Line 255 and 227: authors probably mean “advanced fibrosis”.

Answer 5: Thank you for your suggestion. We mean “advanced fibrosis” and amend the word.

Question 6: Line 249: The data availability statement could be limited to the second period. (i.e.: “Data available on request due to restrictions eg privacy or ethical.”  can be removed).

Answer 6: Thank you for your advice. We removed the description of “Data available on request due to restrictions eg privacy or ethical.”

Reference

1. Blomdahl, J., P. Nasr, M. Ekstedt, and S. Kechagias. "Moderate Alcohol Consumption Is Associated with Advanced Fibrosis in Non-Alcoholic Fatty Liver Disease and Shows a Synergistic Effect with Type 2 Diabetes Mellitus." Metabolism 115 (2021): 154439.

2. McPherson, S., T. Hardy, J. F. Dufour, S. Petta, M. Romero-Gomez, M. Allison, C. P. Oliveira, S. Francque, L. Van Gaal, J. M. Schattenberg, D. Tiniakos, A. Burt, E. Bugianesi, V. Ratziu, C. P. Day, and Q. M. Anstee. "Age as a Confounding Factor for the Accurate Non-Invasive Diagnosis of Advanced Nafld Fibrosis." Am J Gastroenterol 112, no. 5 (2017): 740-51.

Reviewer 2 Report

In this manuscript, authors reported that hyperuricemia was significantly associated with advanced liver fibrosis defined by increased FIB-4 index, especially in non-type 2 diabetic group, in a single center cross sectional study. The subject of study seems to be interesting. However, there are some major concerns in this study. The reviewer’s comments are described as follows.

1. In this study, the definition of hyperuricemia appears to be unclear. Authors must describe the cut-off value of serum uric acid level for hyperuricemia. In addition, whether subjects taking anti-hyperuricemic drugs were included in hyperuricemia group must be explained.

2. In Table 1, serum uric acid level must be added. In addition, serum uric acid levels in diabetic patients can be significantly influenced by renal dysfunction. Markers for renal function including blood urea nitrogen, serum creatinine, and estimated GFR must be included in Table 1 and logistic regression models.

3. According to the definition of type 2 diabetes in this study, subjects with current use of anti-diabetic agents could be included in diabetic group. Subject taking antihypertensive or lipid lowering agents might be also included in this study. Since SGLT2 inhibitors or fibrates are known to prevent the progression of NAFLD, medications should be considered as important confounding factors. Use of these medications and anti-hyperuricemic agents should be included in the analyses.

4. In Table 1, authors described p <0.0001 in advanced fibrosis. However, which grade (low, intermediate, or high) was significant remained unclear. Since the trend of percentage for hyperuricemia in low group and high group appears to be opposite, p value should be clearly presented.

5. In Table 2, how authors adjusted the potential risk factors in each model was never described. Authors must explain the differences among model 1, 2 and 3 in table ledged in detail.

6. In Discussion, authors should explain why exercise was a significant risk factor for advanced liver fibrosis in non-diabetic subjects.

7. The term liver or hepatic fibrosis should be included in the title.

Author Response

Question 1: In this study, the definition of hyperuricemia appears to be unclear. Authors must describe the cut-off value of serum uric acid level for hyperuricemia. In addition, whether subjects taking anti-hyperuricemic drugs were included in hyperuricemia group must be explained.

Answer 1: Thank you for your advice. We added the definition of hyperuricemia definition with cut-off value. Past history of hyperuricemia and anti-hyperuricemia medication were included in our questionnaire. We excluded subject using anti-hyperuricemic agents due to possible association between medications and advanced fibrosis. Therefore, we amended the manuscript as follows:

Line 74-82:

2.1 Subjects

The exclusion criteria were excessive alcohol use (> 10 g of alcohol daily for women and > 20 g for men), previous history of chronic liver diseases (including chronic hepatitis, autoimmune, drug-induced, vascular and inherited hemochromatosis, and Wilson disease). “Subjects using anti-hyperuricemic agents were excluded from our study.” Furthermore, we excluded those who were age equal to or older than 65 years.

Line 110-113:

2.3 Definition of hyperuricemia

Hyperuricemia was defined as serum uric acid > 7.6 mg/dL in men and >6.6 mg/dL in women. Furthermore, subjects with past history of hyperuricemia and taking anti-hyperuricemic medication would be classified as hyperuricemia group.

Line 118-121:

2.5 Diagnosis of chronic kideny disease

The diagnostic criteria for chronic kidney disease(CKD) were based on 1) history of CKD in the questionnaire, 2) estimated glomerular filtration rate(GFR) less than 60 ml/min/1.73m2.

Question 2: In Table 1, serum uric acid level must be added. In addition, serum uric acid levels in diabetic patients can be significantly influenced by renal dysfunction. Markers for renal function including blood urea nitrogen, serum creatinine, and estimated GFR must be included in Table 1 and logistic regression models.

Answer 2: We appreciated your comments. We add serum uric acid level in each group in Table 1 (5.5±1.1 mg/dL in normal group, 8.3±1.0 mg/dL in hyperuricemia group). Indeed, renal function contributed to the clearance of serum uric acid and should be considered into our analysis. We added blood urea nitrogen, serum creatinine and estimated GFR in the Table 1. Furthermore, we also included chronic kidney disease (CKD), which was defined as eGFR<60/ml/min/1.732 or past history of CKD in our questionnaire, in the table 1. For the consideration of renal clearance of uric acid, we further incorporated CKD as a confounding factor in our logistic regression model in Table 2, 3 4.

Question 3: According to the definition of type 2 diabetes in this study, subjects with current use of anti-diabetic agents could be included in diabetic group. Subject taking antihypertensive or lipid lowering agents might be also included in this study. Since SGLT2 inhibitors or fibrates are known to prevent the progression of NAFLD, medications should be considered as important confounding factors. Use of these medications and anti-hyperuricemic agents should be included in the analyses.

Answer 3: We really appreciated your comments. We reviewed our questionnaire, and the question was about use of anti-dyslipidemia medications but not use of fibrates. Because most of anti-dyslipidemia medications in our country means statin. We cannot actually know when subjects use fibrates instead of statin. We would descript it into the limitation.

    Besides, the study was conducted between the 2006 to 2008, and SGLT2 inhibitor was not used in our country. Thus, there might be no confounding effect of SGLT2 inhibitor in this study.

    In our calculation, 1.9% of subjects use anti-hyperuricemic agents. Because using anti-hyperuricemic agents would lower the SUA level and might misclassify subject into normal or hyperuricemia group. We exclude subjects with use of anti-hyperuricemic agents as one of exclusion criteria, which is described in answer 1 above.

Line 864-867:

“Fourth, those who used fibrates for hypertriglycemia might improve advanced fibrosis status. However, we only recorded those who use anti-lipid drugs, which indicated statin in most case. Thus, we cannot adjust the effect of fibrates in our study.”

Question 4: In Table 1, authors described p <0.0001 in advanced fibrosis. However, which grade (low, intermediate, or high) was significant remained unclear. Since the trend of percentage for hyperuricemia in low group and high group appears to be opposite, p value should be clearly presented.

Answer 4: Thank you for your advice. In view of advanced fibrosis in table 1, it is a 3x2 table. We use chi-square test to demonstrate the statistically significant difference between each number. We can find that hyperuricemia group has higher percentage of high risk advance fibrosis than normal group (2.5% v.s. 0.6%). However, it might be confounded by other factors, such as renal function, age. Thus, we use multivariate logistic regression to clarify whether this significant remain after adjusting for confounders.

Question 5: In Table 2, how authors adjusted the potential risk factors in each model was never described. Authors must explain the differences among model 1, 2 and 3 in table legend in detail.

Answer 5: We appreciate your advice. We add the description of variables we adjusted in the statistical analysis of method and legend of table 2 as follow:

Line 134-139:

2.7 Statistical analysis

…We checked risk factors for developing advanced fibrosis with sex, BMI, T2DM, CKD, exercise, smoking and drinking habits. “Model 1 showed the crude odds ratio of advanced fibrosis between two groups. Model 2 added the age, sex, BMI, T2DM and CKD of the participants as confounding factors, while model 3 furtherly included exercise habit and social factors, including alcohol intake and smoking.”

Question 6: In Discussion, authors should explain why exercise was a significant risk factor for advanced liver fibrosis in non-diabetic subjects.

Answer 6: Thank you for your advice. In our table 4 demonstrated exercise was a significant risk factor for advanced liver fibrosis in non-diabetic subjects (OR, 1.92; 95% OR, 1.55-2.37). We discussed this result in the discussion part as follow:

Line 774-738

“Exercise improved hepatic steatosis, underlying metabolic abnormalities and liver fibrosis.[38-40] Exercise alone improved intrahepatic triglyceride.[41] Furthermore, exercise combined with dietary intervention improves serum levels of liver enzymes and liver fat or histology.[39] However, in our table 4 demonstrated exercise was a significant risk factor for advanced liver fibrosis in non-diabetic subjects (OR, 1.92; 95% OR, 1.55-2.37). Whether this result is really clinically significant or just statistically significant is not explored. This result might result from possible interaction of insulin resistance and hepatic steatosis with exercise. Wang et al. demonstrated that physical activity alone can only slightly improve hepatic enzyme levels and intra-hepatic lipid content in non-diabetic patients with NAFLD.[42] Besides, exercise type and intensity might also influence the study result and further study in specific groups were needed.”

Question 7: The term liver or hepatic fibrosis should be included in the title.

Answer 7: Thank you for your advice. We modified our title with the term of hepatic fibrosis as following:

Title:

“Clinical and Metabolic Characteristic of Hyperuricemia with Risk of Hepatic Fibrosis: a Cross-Sectional Study”

Reference

1. Whitsett, M., and L. B. VanWagner. "Physical Activity as a Treatment of Non-Alcoholic Fatty Liver Disease: A Systematic Review." World J Hepatol 7, no. 16 (2015): 2041-52.

2. Katsagoni, C. N., M. Georgoulis, G. V. Papatheodoridis, D. B. Panagiotakos, and M. D. Kontogianni. "Effects of Lifestyle Interventions on Clinical Characteristics of Patients with Non-Alcoholic Fatty Liver Disease: A Meta-Analysis." Metabolism 68 (2017): 119-32.

3. Oh, S., R. So, T. Shida, T. Matsuo, B. Kim, K. Akiyama, T. Isobe, Y. Okamoto, K. Tanaka, and J. Shoda. "High-Intensity Aerobic Exercise Improves Both Hepatic Fat Content and Stiffness in Sedentary Obese Men with Nonalcoholic Fatty Liver Disease." Sci Rep 7 (2017): 43029.

4. Golabi, P., C. T. Locklear, P. Austin, S. Afdhal, M. Byrns, L. Gerber, and Z. M. Younossi. "Effectiveness of Exercise in Hepatic Fat Mobilization in Non-Alcoholic Fatty Liver Disease: Systematic Review." World J Gastroenterol 22, no. 27 (2016): 6318-27.

5. Wang, S. T., J. Zheng, H. W. Peng, X. L. Cai, X. T. Pan, H. Q. Li, Q. Z. Hong, and X. E. Peng. "Physical Activity Intervention for Non-Diabetic Patients with Non-Alcoholic Fatty Liver Disease: A Meta-Analysis of Randomized Controlled Trials." BMC Gastroenterol 20, no. 1 (2020): 66.

Round 2

Reviewer 2 Report

Authors have successfully addressed the reviewer's concerns. There are no more comments.